# Catastrophic costs for tuberculosis patients in India: Impact of methodological choices

Susmita Chatterjee[1,2,3]*, Palash Das[1], Guy Stallworthy[4], Gayatri Bhambure[1], Radha Munje[5], Anna Vassall[6]

1 Research Department, George Institute for Global Health, New Delhi, India, 2 Department of Medicine, University of New South Wales, Kensington, New South Wales, Australia, 3 Prasanna School of Public Health, Manipal Academy of Higher Education, Manipal, Karnataka, India, 4 Bill & Melinda Gates Foundation, Global Health Division, Seattle, Washington State, United States of America, 5 Department of Respiratory Medicine, Indira Gandhi Government Medical College, Nagpur, Maharashtra, India, 6 Department of Global Health and Development, London School of Hygiene and Tropical Medicine, London, United Kingdom

* schatterjee@georgeinstitute.org.in

**Data Availability Statement:** All deidentified data underlying the findings in this article are available as Supporting information files uploaded with this paper.

## Abstract

As financial risk protection is one of the goals towards universal health coverage, detailed information on costs, catastrophic costs and other economic consequences related to any disease are required for designing social protection measures. End Tuberculosis (TB) Strategy set a milestone of achieving zero catastrophic cost by 2020. However, a recent literature review noted that 7%-32% TB affected households in India faced catastrophic cost. Studies included in the review were small scale cross-sectional. We followed a cohort of 1482 notified drug-susceptible TB patients from four states in India: Assam, Maharashtra, Tamil Nadu, and West Bengal to provide a comprehensive picture of economic burden associated with TB treatment. Treatment cost was calculated using World Health Organization guidelines on TB patient cost survey and both human capital and output approaches of indirect cost (time, productivity, and income loss related to an episode) calculation were used to provide the range of households currently facing catastrophic cost. Depending on choice of indirect cost calculation method, 30%-61% study participants faced catastrophic cost. For over half of them, costs became catastrophic even before starting TB treatment as there was average 7–9 weeks delay from symptom onset to treatment initiation which was double the generally accepted delay of 4 weeks. During that period, they made average 8–11 visits to different providers and spent money on consultations, drugs, tests, and travel. Following the largest cohort of drug-susceptible TB patients till date, the study concluded that a significant proportion of study participants faced catastrophic cost and the proportion was much higher when income loss was considered as indirect cost calculation method. Therefore, ensuring uninterrupted livelihood during TB treatment is an absolute necessity. Another reason of high cost was the delay in diagnosis and costs incurred during pre-diagnosis period. This delay and consequently, economic burden, can be reduced by both supply side (intense private sector engagement, rapid diagnosis) and demand side (community engagement) initiatives. Reimbursement of expenses incurred before treatment initiation could be used as short-term measure for mitigating financial hardship.

**Funding:** This work was supported by the DBT/ Welcome Trust India Alliance Clinical and Public Health Intermediate Fellowship [grant number IA/ CPHI/17/1/503339, to SC]. The funder had no role in study design, data collection and analysis, decision to publish, or preparation of the manuscript.

**Competing interests:** SC is an Editorial Board member of this journal. Other authors have no competing interest to declare.

## Introduction

Catastrophic cost, defined in health financing literature as out-of-pocket expenses above a certain threshold proportion of household income, is the reason of impoverishment for millions of people in the world [1]. In tuberculosis (TB), catastrophic cost is defined as total treatment cost (direct + indirect cost) ≥20% pre-TB annual household income as adverse outcomes were strongly associated with this threshold [2, 3]. Because indirect cost, i.e., time, income and productivity loss associated with an episode of TB, contributes a significant proportion of total treatment cost, it is included in the catastrophic cost definition along with out-of-pocket expenses [4].

India has the highest TB burden in the world with reported incidence of 1.93 million in 2021 [5]. The government of India policy is to provide "free" diagnosis and treatment to all registered TB cases; these include basic diagnostic tests (most common tests are sputum smear microscopy, chest x-ray and rapid molecular tests) and anti-TB medications. Patients pay 'out-of-pocket' for several other components such as consultation fees, non-TB drugs, other diagnostic tests (for example, computed tomography scan, magnetic resonance imagining, biopsy), travel expenses in pre-treatment period (i.e., from symptom onset to treatment initiation) and travel expenses for picking up TB drugs, additional food / nutritional supplements, management of side effects during treatment period. Studies showed high cost associated with TB treatment in India and 7%-40% drug-susceptible (DS) TB patients reported facing catastrophe using the threshold of ≥20% of pre-TB household annual income [6–11]. However, majority of those were small scale cross-sectional studies the results of which will be difficult to generalize for Indian context.

As financial risk protection is one of the goals towards universal health coverage, detailed information on costs, catastrophic costs and other economic consequences related to the disease are required for designing social protection measures. Measurement of catastrophic cost related to TB for a representative population of India will help the government to address the demand side cost barriers, and its mitigation strategies. Further, this will be an important indicator towards policy decisions to achieve zero catastrophic cost goal which was supposed to be achieved by 2020 as envisioned in End TB strategy [2].

In this paper, we provided a comprehensive picture of economic burden associated with TB treatment in India. We reported the proportion of TB affected households that faced catastrophic cost using different approaches of indirect cost.

## Materials and methods

### Study design

We followed a cohort of 1482 notified patients with drug-susceptible TB (DS-TB) from the general population (529 patients) as well as from two 'high-risk groups' as defined in the National Strategic Plan for TB: 2017–2025 [12]–patients living in urban slum areas (526 patients) and tea garden areas (427 patients). As the general population can also include urban slum-dwellers, participants from urban areas were sampled ensuring the general population as a non-overlapping group. Further, general population group had participants from both urban and rural areas. All study participants were sampled from 118 TB units and 182 tea gardens across 16 districts of four states in India and were representative of their respective groups. One TB unit covers an average 200,000 (range 150,000–250,000) population in rural and urban areas. Patients with TB from 'high-risk groups' were chosen as their treatment seeking behaviour, treatment cost, proportion facing catastrophic cost and treatment outcomes have never been examined in the Indian context and may be useful for future policies towards TB

elimination in the country. The study states (Assam, Maharashtra, Tamil Nadu, and West Bengal) were chosen as they represented different levels of development and covered different regions of the country. Further, Assam (northeast India), Tamil Nadu (south), and West Bengal (east) are the states where most tea gardens are located, while Maharashtra (west), Tamil Nadu and West Bengal are among the top five states in terms of urban slum population in the country. Methods of state selection, details of 'high-risk groups', and sample size calculation for the study are presented elsewhere [11].

## Study participants, interviews, and instruments

Adult (18 years and above) patients registered for TB treatment at government health facilities, at intensive phase (IP) of treatment (i.e., at first two months of TB treatment) during the visit of the study team and gave written informed consent to participate in the study were interviewed at IP, at 5–6 months of treatment and about one-year post-treatment. In this paper, we restricted our analyses from TB symptom onset till the end of treatment to examine the economic burden associated with a TB episode and did not consider costs incurred in the post-treatment phase.

The questionnaires were adapted from the World Health Organization (WHO) guideline for TB patient cost surveys [2]. All interviews covered time and out-of-pocket expenses on outpatient visits, hospitalizations, drug pick-ups / directly observed treatment, medical follow-ups related to the TB episode along with expenses on additional food/supplements. Socioeconomic characteristics, employment and income of the patient and household income, ownership of assets and coping strategies (savings withdrawn / borrowing / sale/mortgage of personal belongings) were also covered. Study participants were interviewed between March 2019 to September 2022 by 12 trained researchers under close supervision by two core study team members.

## Cost calculation methods

Total treatment cost of a TB episode was the sum of direct and indirect cost incurred from the onset of TB symptoms until treatment completion. Direct medical costs included out-of-pocket payments for consultation fees, bed charges, diagnostics tests, medicines, and surgeries, while direct non-medical costs included payments for travel expenses and food during outpatient visits / hospitalization / drug pick-up / directly observed treatment, additional food/supplements, and any other payments. Indirect costs are calculated using two approaches: human capital approach (HCA) and output approach (OA) [2]. In HCA, indirect cost is calculated by multiplying reported time use while seeking and receiving care during the TB episode by either minimum wage rate or reported hourly wage rate of the patient. Therefore, HCA measures the lost productivity but does not capture indirect cost due to reduced productivity as it multiplies the time spent while seeking and receiving care with the wage rate. Minimum wage rate is used from an equity lens where all time spent is valued using same wage rate. It can also be used to avoid any reporting bias in actual wage rate. In OA, indirect cost is calculated as a difference between self-reported household income before TB and during TB treatment [2]. Therefore, in OA, reduced productivity in the form of income loss is captured but it misses the indirect costs associated with the time spent by the patients and guardians who are not involved in paid services (such as housewives, student). As two different approaches provide two different estimates, in this study, we calculated Indirect cost using both HCA and OA methods. In HCA, we used both minimum wage and reported wage to provide the ranges of indirect cost. Therefore, three different approaches were used to calculate catastrophic cost: (1) value of hours spent by patient and guardian for seeking and receiving care were calculated

using minimum wage rates of respective study states [13]–HCA1; (2) value of hours spent by employed patients were estimated by using reported hourly wage rate of the patients, while the value of time spent by others was estimated using minimum wage rates–HCA2; and (3) difference in self-reported household income during pre-TB period and during treatment–OA [2]. We used the following formula to estimate the indirect cost using OA:

(Pre-TB household income–household income in IP)*time difference between pre and IP interview + (household income in IP–household income at end of interview)* time difference between IP and end of treatment interview.

Catastrophic cost was defined as total TB treatment cost (direct + indirect cost) ≥20% pre-TB annual household income [2]. Earlier Indian studies used annual pre-TB household income as denominator of catastrophic cost calculation [6–11]. However, household expenditure is a better measure of permanent income of the household as it generally stays stable over time [14, 15]. Therefore, apart from using pre-TB household income as denominator for catastrophic cost calculation, we also used pre-TB household expenditure as denominator [2].

Details of the approaches used for calculating indirect costs and catastrophic costs are presented in Table 1. All costs are presented in 2022 Indian Rupees (INR) (1 US$ = INR 78.5344).

To determine the likelihood of facing catastrophic cost for our study participants, we ran binary logistic regressions for each group and all groups combined. Age, sex, education, pre-TB annual household income, wealth quintile, health insurance, type of TB, delay from symptom onset to treatment initiation, direct cost of TB treatment and residential status were the independent variables. Descriptive statistics of the predictor variables are presented in (S1 Table).

## Coping strategies

Studies reported that presence of coping strategies can be an indicator of catastrophe for the TB affected households [16]. We therefore asked study participants during both IP and end of

**Table 1. Approaches for calculating catastrophic cost.**

| Approaches | Indirect cost calculation method | Denominator of catastrophic cost |
|---|---|---|
| HCA1 | Human capital approach (using minimum wage rate for all) | Self-reported pre-TB household income |
| HCA2 | Human capital approach (using reported patient wage for employed patient and minimum wage rate for others) | Self-reported pre-TB household income |
| OA1 | Output approach (difference between self-reported household income before TB and during treatment) | Self-reported pre-TB household income |
| HCA3 | Human capital approach (using minimum wage rate for all) | Self-reported pre-TB household expenditure |
| HCA4 | Human capital approach (using reported patient wage for employed patient and minimum wage rate for others) | Self-reported pre-TB household expenditure |
| OA2 | Output approach (difference between self-reported household income before TB and during treatment) | Self-reported pre-TB household expenditure |

Notes: HCA1: Human capital approach where hours spent was calculated using minimum wage rate for all and household income as denominator; HCA2: Human capital approach where hours spent was calculated using combination of patient wage and minimum wage and household income as denominator; OA1: Output approach with household income as denominator; HCA3: Human capital approach where hours spent was calculated using minimum wage rate for all and household expenditure as denominator; HCA4: Human capital approach where hours spent was calculated using combination of patient wage and minimum wage and household expenditure as denominator; OA2: Output approach with household expenditure as denominator.

treatment interviews to report various coping strategies for managing the cost of the disease and other economic consequences such as reducing consumption, other household members started working, pulling out children from school / private tuition, running up account in shop, etc.

### Sensitivity analyses

The main analyses were presented for all study participants (1482). We also estimated treatment cost by excluding participants who were missed during follow-up interviews at the end of treatment to examine the changes in cost and catastrophic cost.

**Ethics approval and consent to participate.** The study was approved by the Institutional Ethics Committee of the George Institute for Global Health (014/2018). Written informed consent was obtained from each study participant before starting the interview.

## Results

Details of characteristics of study participants, health seeking behaviour, treatment outcome and income are provided in Table 2. Characteristics of study participants interviewed at the end of treatment was similar as IP.

Male patients dominated in our study cohort (59%-66%) and 64%-78% of all study participants were in the age groups of 18–44 years. Majority participants among general population (63%) and urban slum dwellers (69%) had at least secondary education, however, only 26% participants among tea garden families had similar education. Significant proportion of study participants were employed (61%-67%) and 42%-45% were the main earners of the family before TB. Insurance coverage among study participants was poor (21%-34%). Majority participants (59%-75%) were covered through state health insurance schemes followed by central government health insurance schemes (11%-26%). Bacteriologically confirmed TB was the commonest form for all types of participants (56%-61%) while 25%-31% study participants had extrapulmonary (EP) TB (TB involving organs other than the lungs).

### Health seeking behaviour

Average delay from the onset of symptom to treatment initiation ranged from 7 weeks (participants from tea garden families) to 9 weeks (urban slum dwellers). Majority study participants among general population and urban slum dwellers (72%-75%) first contacted a private provider after symptom onset and they made on average 8–11 visits to different providers before they were diagnosed with TB. Among tea garden residents, only 31% study participants' first choice was a private provider after symptom onset.

### Treatment outcome

Adverse treatment outcomes defined as death, relapse cases, discontinuation of treatment ranged from 12%-14% among study participants. Discontinuation of treatment for two consecutive months was the highest among participants from urban slums (5%) while deaths were the highest for participants among tea garden families (8%).

### Income

Pre-TB monthly household income was higher for participants from the general population (INR 20,381), and urban slum dwellers (INR 16,984) as compared to participants from tea gardens (INR 9,560). For all groups, patient and household income decreased during IP of

**Table 2. Characteristics of the study participants.**

| | General population (N = 529) | Urban slum dwellers (N = 526) | Tea garden families (N = 427) |
|---|---|---|---|
| **Demographic characteristics** | | | |
| **Sex** | | | |
| Female | 182 (34%) | 218 (41%) | 173 (41%) |
| Male | 347 (66%) | 308 (59%) | 254 (60%) |
| **Age** | | | |
| 18–24 years | 97 (18%) | 115 (22%) | 114 (27%) |
| 25–34 years | 127 (24%) | 117 (22%) | 130 (30%) |
| 35–44 years | 116 (22%) | 105 (20%) | 90 (21%) |
| 45–54 years | 89 (17%) | 95 (18%) | 47 (11%) |
| 55–64 years | 66 (13%) | 73 (14%) | 36 (8%) |
| 65 years and above | 34 (6%) | 21 (4%) | 10 (2%) |
| **Education** | | | |
| No education | 88 (17%) | 81 (15%) | 170 (40%) |
| Up to primary education | 110 (21%) | 84 (16%) | 145 (34%) |
| Secondary education and above | 331 (63%) | 361 (69%) | 112 (26%) |
| **Employment status before TB** | | | |
| Employed | 338 (64%) | 322 (61%) | 285 (67%) |
| Unemployed | 191 (36%) | 204 (39%) | 142 (33%) |
| **Insurance status** | | | |
| Yes | 149 (28%) | 109 (21%) | 146 (34%) |
| No | 375 (71%) | 413 (79%) | 281 (66%) |
| Don't know | 5 (1%) | 4 (1%) | 0 (0%) |
| **Median household size (min-max)** | 4 (1–15) | 4 (1–17) | 5 (1–12) |
| **Clinical characteristics and treatment seeking behaviour** | | | |
| **Type of TB** | | | |
| Bacteriologically confirmed pulmonary TB | 298 (56%) | 320 (61%) | 256 (60%) |
| Clinically diagnosed pulmonary TB | 66 (13%) | 55 (11%) | 63 (15%) |
| Extrapulmonary TB | 165 (31%) | 151 (29%) | 108 (25%) |
| **Adverse outcomes** | | | |
| Death | 25 (5%) | 23 (4%) | 32 (8%) |
| Discontinuation of treatment | 22 (4%) | 28 (5%) | 7 (2%) |
| Relapse | 20 (4%) | 18 (3%) | 16 (4%) |
| **Average delay from symptom onset to treatment initiation—weeks (SD)*** | 9 (9) | 9 (9) | 7 (8) |
| **First point of contact after symptom onset** | | | |
| Government provider / tea garden hospital | 151 (29%) | 131 (25%) | 294 (69%) |
| Private provider | 378 (72%) | 395 (75%) | 133 (31%) |
| **Average number of visits before diagnosis (SD)** | 10 (8) | 11 (10) | 8 (5) |
| **Hospitalization** | | | |
| Total hospitalized | 155 (29%) | 133 (25%) | 129 (30%) |
| Government / tea garden hospitals | 105 (68%) | 92 (69%) | 112 (87%) |
| Private hospitals | 46 (30%) | 35 (26%) | 8 (6%) |
| Others** | 4 (3%) | 6 (5%) | 9 (7%) |
| **Treatment support** | | | |
| Self-administered | 453 (86%) | 417 (79%) | 236 (55%) |
| Directly observed therapy | 76 (14%) | 109 (21%) | 191 (45%) |
| **Monthly patient and household income—2022 INR—Mean (SD)** | | | |

*(Continued)*

**Table 2.** (Continued)

| | General population (N = 529) | Urban slum dwellers (N = 526) | Tea garden families (N = 427) |
|---|---|---|---|
| Patient as the main earner of the family before TB (%) | 239 (45%) | 229 (44%) | 177 (42%) |
| Patient income before TB | 8575 (12037) | 7179 (9033) | 3323 (3942) |
| Household income before TB | 20381 (18526) | 16984 (14174) | 9560 (6707) |
| Patient income at intensive phase of treatment | 4389 (10594) | 2777 (5877) | 868 (3225) |
| Household income at intensive phase of treatment | 16199 (18420) | 12883 (12985) | 6981 (6614) |
| Patient income during the end of treatment interview*** | 6002 (11453) | 4092 (6883) | 1217 (3350) |
| Household income during the end of treatment interview*** | 16992 (18550) | 13334 (10902) | 7051 (6130) |
| **Monthly household expenditure before TB (2022 INR)—mean (SD)** | 14947 (10850) | 15206 (8393) | 6981 (3688) |
| **Residential status** | | | |
| Urban | 277 (52%) | 526 (100%) | 0 (0%) |
| Rural | 252 (48%) | 0 (0%) | 427 (100%) |

Notes:

*SD = standard deviation;

**Others include trust / charitable hospitals, public private partnership hospitals;

*** patient and household income during the end of treatment interview was for 497 participants among general population, 499 among urban slum dwellers and 398 in tea garden areas; 1 US$ = INR 78.5344

treatment as compared to pre-TB income and improved during rest of the treatment period, however, it did not reach the baseline.

## Reasons of missed follow-up interviews

We missed 6% study participants among general population during the end of treatment interviews and the reasons were death (2.6%), refusal (1.7%), not available in given address / over phone (1.1%) and others (0.6%). Major reasons of missing interviews of 28 participants among urban slum dwellers were not available in given address / over phone (2.5%), death (1.3%) and refusal (1.3%). We missed 29 participants (6.8%) during end of treatment interviews in tea garden areas because of death (5.2%), not available in given address / over phone (1.2%) and others (0.5%).

## Treatment cost

Total TB treatment cost using HCA1 was INR 29,960 (standard deviation–SD 58,781) for participants in tea garden areas, INR 30,782 (SD 44061) for urban slum dwellers and INR 32,829 (SD 49,889) for participants in general population (Table 3). Using HCA2, total treatment cost varied from INR 22,981 (SD 33,448) to INR 34,315 (SD 43,914). Total treatment cost using OA ranged from INR 30,347 (SD 30,612) for tea garden families to INR 57,992 (SD 80,340) for urban slum dwellers to INR 61,181 (SD 65,348) for participants in general population. Using HCA1 and HCA2 methods of indirect cost calculation, 58%-59% of total cost was incurred during the pre-treatment phase for participants among urban slum dwellers and general population respectively. Direct cost was higher than indirect cost in all phases for these two groups. During pre-treatment phase, major reason of higher direct cost was outpatient visits while during treatment, it was buying nutritional food and supplements. For participants from tea garden areas, 44% cost was incurred in pre-treatment phase and indirect cost dominated during pre- and IP of treatment using HCA1 and HCA2 methods of indirect cost calculation. Reason of higher indirect cost among participants in tea garden areas was higher inpatient days. As

**Table 3. Total TB treatment cost in different phases using different indirect cost calculation methods (2022 Indian Rupee).**

| Treatment Phases | General population (N = 529) | | Urban slum dwellers (N = 526) | | Tea garden families (N = 427) | |
|---|---|---|---|---|---|---|
| **Pre-treatment phase** | | | | | | |
| Direct cost | 11951.52 (61.5%) | | 9422.77 (53.0%) | | 2830.73 (21.4%) | |
| Indirect cost using HCA 1 | 7489.33 (38.5%) | | 8351.28 (47.0%) | | 10415.60 (78.6%) | |
| *Total cost using HCA 1* | | 19440.85 | | 17774.05 | | 13246.33 |
| Direct cost | 11951.52 (57.7%) | | 9422.77 (52.7%) | | 2830.73 (26.5%) | |
| Indirect cost using HCA 2 | 8765.97 (42.3%) | | 8468.73 (47.3%) | | 7852.75 (73.5%) | |
| *Total cost using HCA 2* | | 20717.49 | | 17891.50 | | 10683.48 |
| Indirect cost using OA* | - - | | - - | | - - | |
| **Intensive phase** | | | | | | |
| Direct cost | 4949.64 (77.7%) | | 3780.79 (68.5%) | | 2171.54 (18.9%) | |
| Indirect cost using HCA 1 | 1418.28 (22.3%) | | 1736.13 (31.5%) | | 9312.39 (81.1%) | |
| *Total cost using HCA 1* | | 6367.92 | | 5516.92 | | 11483.93 |
| Direct cost | 4949.64 (76.7%) | | 3780.79 (68.6%) | | 2171.54 (27.8%) | |
| Indirect cost using HCA 2 | 1505.32 (23.3%) | | 1732.77 (31.4%) | | 5645.90 (72.2%) | |
| *Total cost using HCA 2* | | 6454.96 | | 5513.56 | | 7817.44 |
| Direct cost | 4949.64 (22.3%) | | 3780.79 (15.9%) | | 2171.54 (17.9%) | |
| Indirect cost using OA | 17230.88 (77.7%) | | 20065.57 (84.1%) | | 9983.34 (82.1%) | |
| *Total cost using OA* | | 22180.52 | | 23846.36 | | 12154.88 |
| **Continuation phase** | | | | | | |
| Direct cost | 5796.28 (82.6%) | | 5074.33 (67.7%) | | 2842.71 (54.4%) | |
| Indirect cost using HCA 1 | 1223.71 (17.4%) | | 2416.25 (32.3%) | | 2387.46 (45.6%) | |
| *Total cost using HCA 1* | | 7019.99 | | 7490.58 | | 5230.17 |
| Direct cost | 5796.28 (81.2%) | | 5074.33 (68.6%) | | 2842.71 (63.5%) | |
| Indirect cost using HCA 2 | 1346.18 (18.8%) | | 2326.20 (31.4%) | | 1636.97 (36.5%) | |
| *Total cost using HCA 2* | | 7142.46 | | 7400.53 | | 4479.68 |
| Direct cost | 5796.28 (21.4%) | | 5074.33 (20.5%) | | 2842.71 (18.5%) | |
| Indirect cost using OA | 21252.39 (78.6%) | | 19648.38 (79.5%) | | 12518.20 (81.5%) | |
| *Total cost using OA* | | 27048.67 | | 24722.71 | | 15360.92 |
| **Entire illness** | | | | | | |
| Direct cost | 22697.45 (69.1%) | | 18277.89 (59.4%) | | 7844.98 (26.2%) | |
| Indirect cost using HCA 1 | 10131.32 (30.9%) | | 12503.67 (40.6%) | | 22115.45 (73.8%) | |
| *Total cost using HCA 1* | | 32828.77 | | 30781.56 | | 29960.43 |
| Direct cost | 22697.45 (66.1%) | | 18277.89 (59.3%) | | 7844.98 (34.1%) | |
| Indirect cost using HCA 2 | 11617.47 (33.9%) | | 12527.70 (40.7%) | | 15135.62 (65.9%) | |
| *Total cost using HCA 2* | | 34314.92 | | 30805.59 | | 22980.60 |
| Direct cost | 22697.45 (37.1%) | | 18277.89 (31.5%) | | 7844.98 (25.9%) | |
| Indirect cost using OA | 38483.26 (62.9%) | | 39713.95 (68.5%) | | 22501.55 (74.1%) | |
| *Total cost using OA* | | 61180.71 | | 57991.84 | | 30346.53 |

Notes: 1 US$ = INR 78.5344.

* Income loss during pre-treatment phase was included in intensive phase income loss. HCA = human capital approach; OA = output approach

majority patients were admitted at the tea garden hospitals, government hospitals, or trust/charitable hospitals (about 95% of all admitted cases) which provided free of cost care, direct cost was lower than indirect cost. Only three study participants out of 1482 received reimbursements through health insurance for hospitalization. Higher direct cost during treatment period was for buying additional food.

TB treatment cost using HCA1 method of indirect cost calculation was much higher for EPTB patients as compared to pulmonary (P) TB patients (not reported in table). While treatment cost ranged from INR 24,318 (SD 36,158) to INR 27,540 (SD 45,597) for PTB patients, it ranged from INR 33,547 (SD 85,795) to INR 48,320 (SD 63,678) for EPTB patients. For participants with PTB, 46%-50% of total treatment cost was incurred in the pre-treatment phase, while the same was about 68% for EPTB patients among general population and urban slum dwellers. On the other hand, proportion of costs incurred in pre-treatment phase for participants with EPTB among tea garden residents was 32% probably indicating high cost associated with treatment seeking and diagnosis through private providers.

## Catastrophic cost

Figs 1 and 2 show the impact of the three methodological choices (method of valuing time, income or expenditure as denominator, and threshold for catastrophic expenditure) on the proportions found to face catastrophic costs. Using treatment cost ≥20% of pre-TB household annual income as threshold, 30%-61% TB affected households faced catastrophe irrespective of indirect cost calculation methods. This proportion ranged from 28%-64% using pre-TB household expenditure as denominator. Proportion of households with catastrophic cost was the highest using OA, indicating the high level of unemployment and hence, income loss associated with TB. HCA1 and HCA2 resulted in similar proportions of households with catastrophic expenditure among general population and urban slum dwellers. However, this proportion was higher using HCA1 for tea garden families as compared to HCA2 (Figs 1 and 2).

Fig 3 shows the relationship between pre-treatment household income and total TB treatment costs for respondents for each study group, using HCA1 as the indirect cost calculation method as this method puts equal value for all time loss. All those to the right of the 20% line faced catastrophic costs at that threshold. Out of total 1482 study participants, 498 (34%) faced

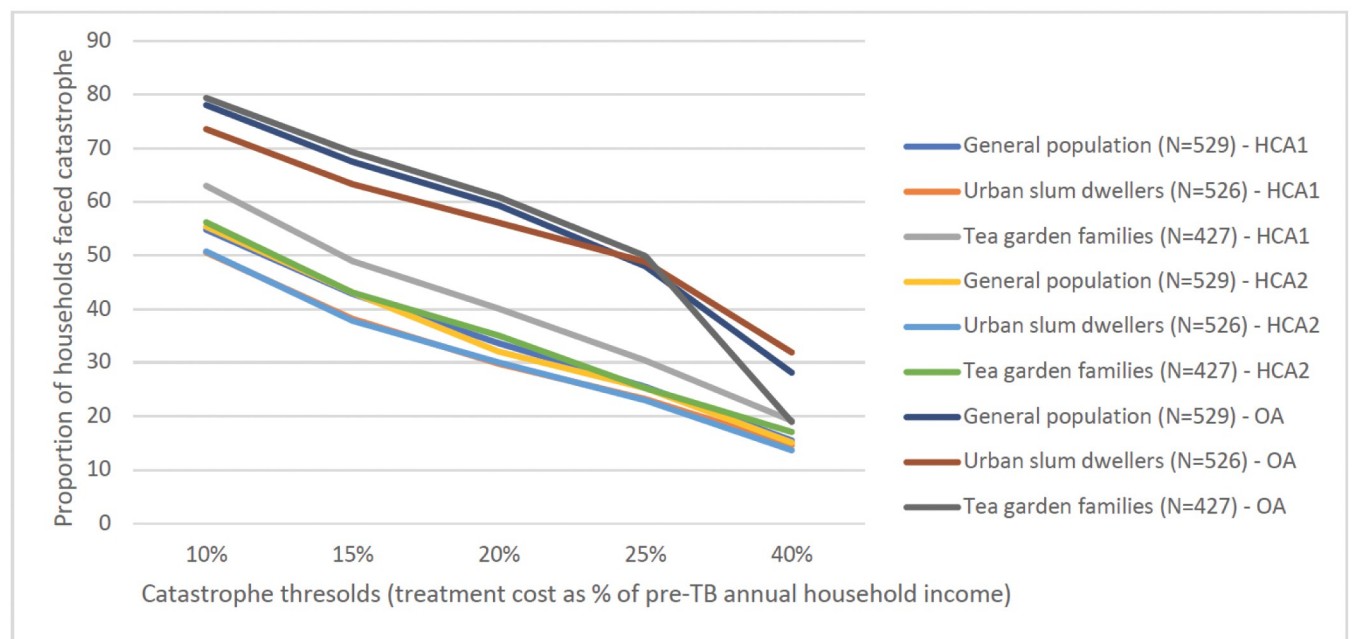

**Fig 1. Proportion of TB-affected households faced catastrophe by indirect cost calculation approach and at various thresholds, using household income as denominator.** Note: HCA1: Human capital approach where hours spent was calculated using minimum wage rate for all; HCA2: Human capital approach where hours spent was calculated using combination of patient wage and minimum wage; OA: Output approach.

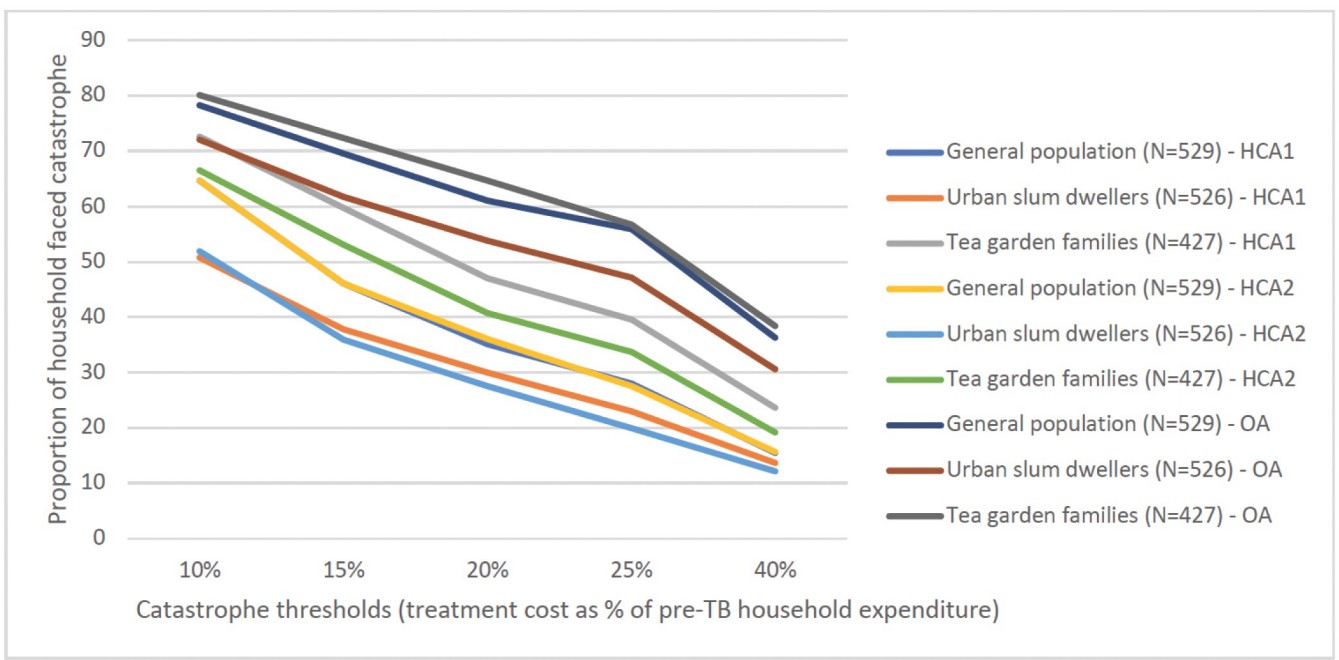

**Fig 2. Proportion of TB-affected households faced catastrophe by indirect cost calculation approach and at various thresholds, using household expenditure as denominator.** Note: HCA1: Human capital approach where hours spent was calculated using minimum wage rate for all; HCA2: Human capital approach where hours spent was calculated using combination of patient wage and minimum wage; OA: Output approach.

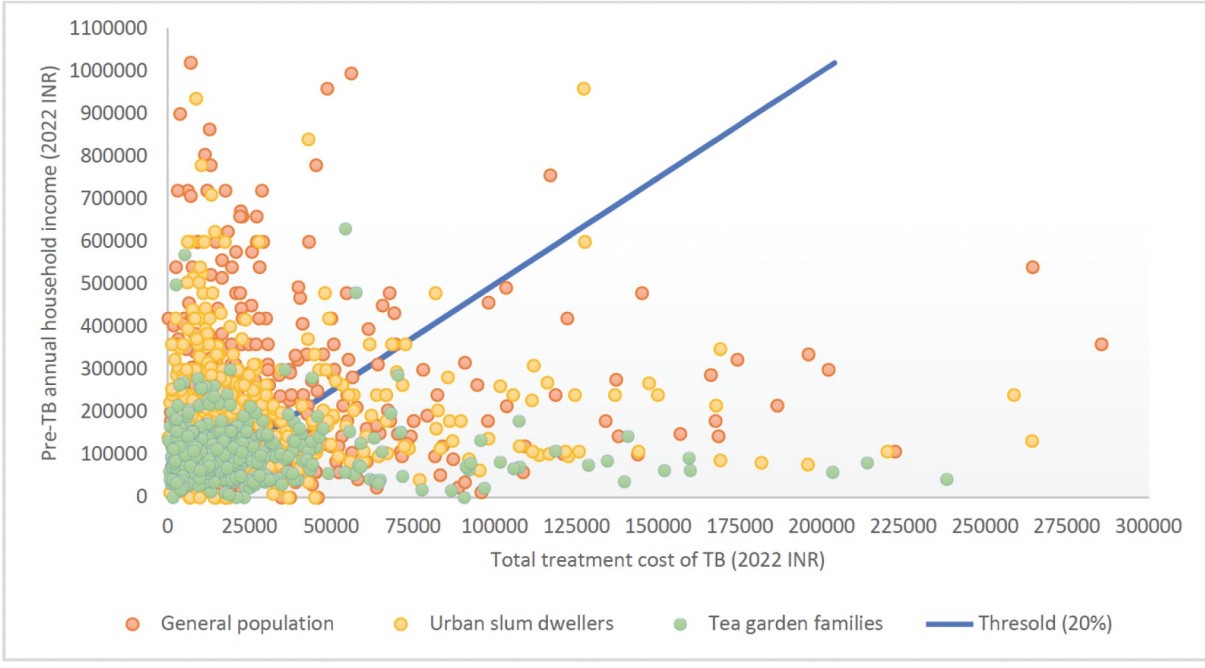

**Fig 3. Catastrophic cost threshold and study participants (N = 1471).** Notes: Total treatment cost was calculated using minimum wage rate to estimate time loss. We excluded 6 study participants from general population, 3 from urban slum dwellers and 2 from tea garden areas as they had either total treatment cost >INR 300,000 or pre-TB annual household income >INR 10,50,000.

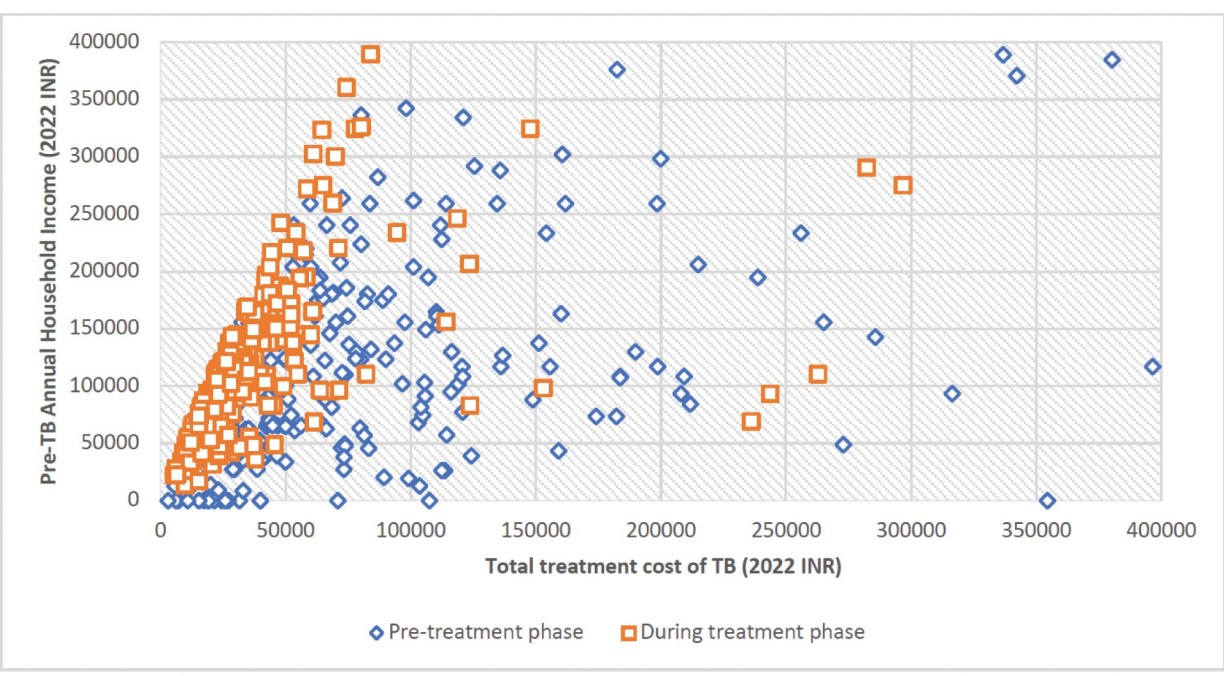

**Fig 4. TB affected households with catastrophic cost at pre-treatment and during treatment phases (N = 1475).** Note: We excluded 7 study participants from this graph as they had either total cost of TB treatment >Indian Rupee 400,000 or pre-treatment annual household income >Indian Rupee 500,000.

catastrophic costs using HCA1, of whom 52% faced it in the pre-treatment phase. Further, costs varied significantly during the pre-treatment phase as compared to treatment phase (Fig 4). There were 18 study participants who had zero household income. This was either because they were old and living using their savings, or they were living alone with no income and their daily expenses were taken care of friends/family members/neighbours who were not part of the household. During their sickness, their guardians took care of treatment expenses, however, as they were not part of the household, their incomes were not considered as household income. Hence, for these study participants, as household income was zero, any cost incurred during TB treatment was catastrophic.

We combined all categories of participants to understand wealth quintile wise distribution of catastrophic cost (Fig 5). As expected, the poorest quintile faced catastrophic cost the most using HCA methods of indirect cost calculation. However, using OA, the middle quintile was more likely to face catastrophic costs. This may be because OA considers the value of lost income, which was higher for those in middle quintiles but not high enough to mitigate costs.

Catastrophic cost was the highest for EPTB patients followed by clinically diagnosed PTB and microbiologically confirmed PTB using HCA1 and HCA2 as indirect cost calculation methods, however, using OA, proportion did not vary by type of TB. But proportion of households faced catastrophic cost using this method was much higher as compared to HCA1 and HCA2, irrespective of types of TB. This indicates that there was significant household income loss because of the disease (Fig 6).

### Predictors of catastrophic cost

We present adjusted odds ratios for all explanatory variables for each group and for various indirect cost calculation approaches in Tables 4 to 6. Unadjusted odds ratios are reported in S2

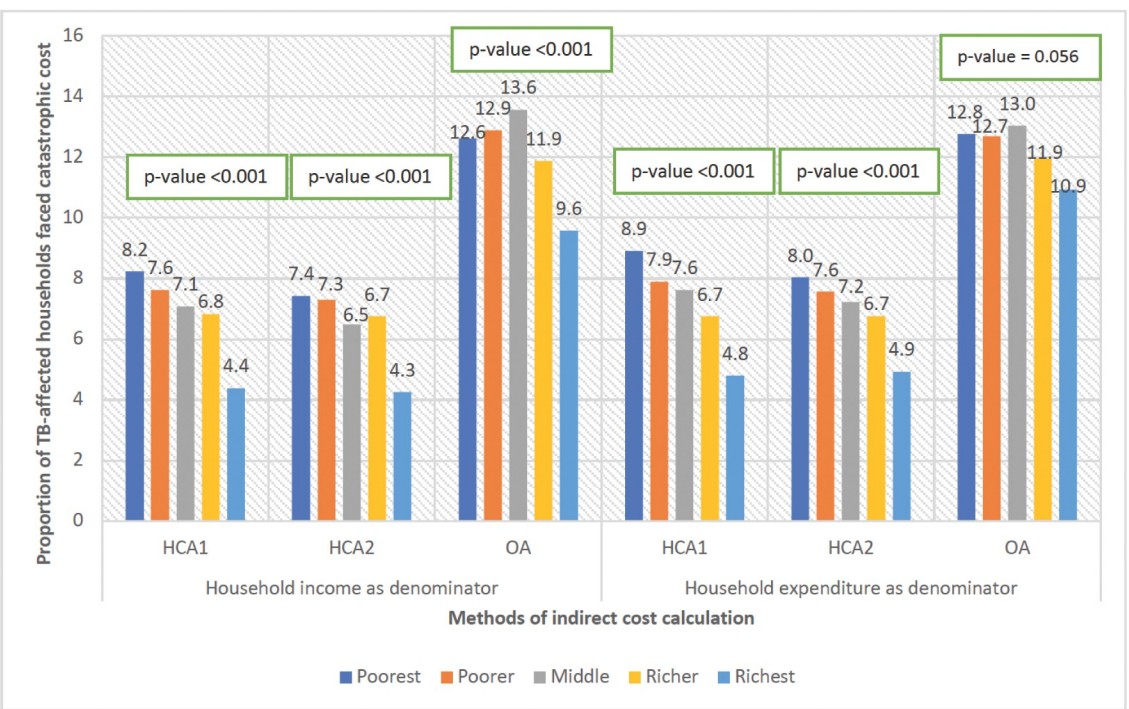

**Fig 5. Proportion of TB-affected households with catastrophic cost, by wealth quintile (N = 1482).** Note: HCA1: Human capital approach where hours spent was calculated using minimum wage rate for all; HCA2: Human capital approach where hours spent was calculated using combination of patient wage and minimum wage; OA: Output approach.

to S4 Tables. Our results showed irrespective of indirect cost calculation method and participant group, lower household income and higher direct cost of TB treatment (i.e., actual money spent for TB treatment) were significantly associated with risk of incurring catastrophic cost. Apart from these, combining all groups, older and female patients, EPTB patients and patients with lower education and in poorest quintiles had higher likelihood of having catastrophic cost using HCA methods of indirect cost calculation.

Using OA as the method of indirect cost calculation, for the combined groups, male patients, patients with lower education and income, higher diagnosis delay and higher direct cost of TB treatment had higher odds of experiencing catastrophic cost.

## Coping strategies and other economic consequences

To manage expenses related to TB, 28%-44% participants borrowed money during IP while 23%-34% borrowed during the rest of the treatment period (Table 7). Selling / mortgage of personal belongings during IP and CP was the highest for the urban slum dwellers (15% and 16% respectively) followed by about 12% in both phases for participants from general population and 10%-14% for participants in tea garden areas. Significant proportion of study participants used several other strategies to mitigate the financial hardship during the illness (Table 7). Even though TB treatment is free for the registered patients, however, as many patients became unemployed and lost income because of the disease, and they spent for buying additional food/nutritional supplements during TB treatment, they continued to borrow / sale /mortgage and used several other strategies throughout TB treatment period.

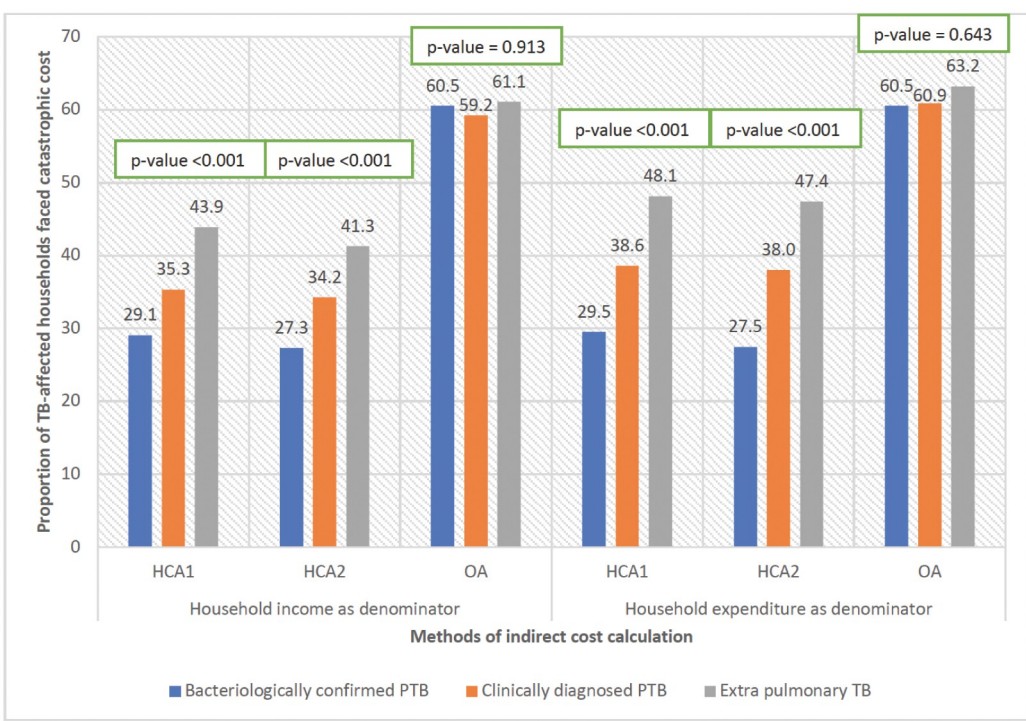

**Fig 6. Proportion of TB-affected households with catastrophic cost, by type of TB (N = 1482).** Note: HCA1: Human capital approach where hours spent was calculated using minimum wage rate for all; HCA2: Human capital approach where hours spent was calculated using combination of patient wage and minimum wage; OA: Output approach.

## Sensitivity analyses

Total treatment cost using HCA1 excluding 32 participants among general population (who were missed during the end of treatment interviews) was INR 33,201 (SD 49,791) as compared to INR 32,829 (SD 49,889) including all participants. Total cost was similar using HCA2 –INR 34,795 (SD 44,005) excluding 32 participants and INR 34,315 (SD 43,914) including all and using OA–INR 61,181 (SD 65,348) and INR 63,175 (SD 65,905) respectively.

Total cost using HCA1 was INR 30,896 (SD 44,525) excluding 28 participants among urban slum dwellers (missed during follow-up interviews) as compared to INR 30,782 (SD 44,061) including them. Using HCA2, total costs were INR 30,942 (SD 45,872) and INR 30,806 (SD 45,364) and using OA, those were INR 59,756 (SD 81,873) and INR 57,992 (SD 80,340) excluding missed participants during follow-up and including them respectively.

Total treatment costs using all indirect cost calculation methods were similar excluding 31 participants who were missed during follow-up interviews in tea garden areas when compared with costs including all.

## Discussion

The study, following a cohort of 1482 notified patients with DS-TB selected from 118 TB units covering both rural and urban areas and 182 tea gardens across 16 districts of four states, provided comprehensive picture of economic burden associated with TB treatment in India. Using TB treatment cost ≥20% of pre-TB annual household income as threshold, our study found that 30%-61% study participants faced catastrophic cost depending on the methods of indirect cost calculation. This information will help the National TB Elimination Programme

**Table 4. Likelihood of incurring catastrophic cost using HCA1 method of indirect cost calculation.**

| | General population (N = 528) | | Urban slum dwellers (N = 526) | | Tea garden families (N = 403) | | All participants (N = 1457) | |
|---|---|---|---|---|---|---|---|---|
| Explanatory variables | Adjusted OR (95% CI) | p-value | Adjusted OR (95% CI) | p-value | Adjusted OR (95% CI) | p-value | Adjusted OR (95% CI) | p-value |
| **Age (in years)** | 1.01 (0.99, 1.03) | 0.18 | 1.01 (0.99, 1.03) | 0.22 | 1.01 (1.00, 1.03) | 0.28 | 1.01 (1.00, 1.02) | 0.05 |
| **Sex** | | | | | | | | |
| Male (Reference) | 1.00 | | 1.00 | | 1.00 | | 1.00 | |
| Female | 1.06 (0.66, 1.70) | 0.80 | 1.59 (0.91, 2.77) | 0.10 | 1.34 (0.85, 2.12) | 0.21 | 1.34 (1.02, 1.76) | 0.04 |
| **Education** | | | | | | | | |
| Up to primary education (Reference) | 1.00 | | 1.00 | | 1.00 | | 1.00 | |
| Secondary education | 0.99 (0.59, 1.67) | 0.98 | 0.63 (0.34, 1.16) | 0.13 | 1.19 (0.64, 2.20) | 0.58 | 0.80 (0.58, 1.10) | 0.17 |
| Higher secondary education and above | 0.48 (0.26, 0.88) | 0.02 | 0.62 (0.28, 1.37) | 0.23 | 0.57 (0.22, 1.48) | 0.24 | 0.61 (0.40, 0.91) | 0.02 |
| **Pre-TB annual household income (Indian Rupee)** | | | | | | | | |
| Less than 100,000 (Reference) | 1.00 | | 1.00 | | 1.00 | | 1.00 | |
| 100,000 –less than 200,000 | 0.13 (0.07, 0.23) | <0.001 | 0.14 (0.08, 0.27) | <0.001 | 0.19 (0.11, 0.31) | <0.001 | 0.16 (0.11, 0.21) | <0.001 |
| 200,000 and above | 0.06 (0.03, 0.12) | <0.001 | 0.03 (0.01, 0.06) | <0.001 | - - - | | 0.07 (0.05, 0.11) | <0.001 |
| **Health insurance** | | | | | | | | |
| Having health insurance (Reference) | 1.00 | | 1.00 | | 1.00 | | 1.00 | |
| Not having health insurance | 1.44 (0.88, 2.34) | 0.15 | 1.03 (0.53, 1.99) | 0.93 | 0.97 (0.60, 1.58) | 0.91 | 1.11 (0.83, 1.50) | 0.48 |
| **Type of TB** | | | | | | | | |
| Pulmonary TB (Reference) | 1.00 | | 1.00 | | 1.00 | | 1.00 | |
| Extrapulmonary TB | 1.54 (0.95, 2.50) | 0.08 | 2.21 (1.23, 3.98) | 0.01 | 1.21 (0.72, 2.03) | 0.48 | 1.68 (1.25, 2.26) | <0.001 |
| **Delay from symptom initiation to treatment** | 1.01 (0.99, 1.03) | 0.41 | 1.01 (0.98, 1.04) | 0.625 | 1.01 (0.98, 1.04) | 0.44 | 1.01 (1.00, 1.03) | 0.08 |
| **Direct cost of TB treatment (Log cost) *** | 2.88 (2.22, 3.74) | <0.001 | 7.37 (4.85, 11.21) | <0.001 | 1.70 (1.41, 2.05) | <0.001 | 2.66 (2.30, 3.06) | <0.001 |
| **Residential status** | | | | | | | | |
| Urban (Reference) | 1.00 | | - - - | | - - - | | 1.00 | |
| Rural | 0.90 (0.57, 1.42) | 0.66 | | | | | 1.27 (0.95, 1.70) | 0.11 |
| **Wealth quintile** | | | | | | | | |
| Poorest (Reference) | - - - | | - - - | | - - - | | 1.00 | |
| Poorer | | | | | | | 0.68 (0.45, 1.02) | 0.06 |
| Middle | | | | | | | 0.64 (0.42, 0.96) | 0.03 |
| Richer | | | | | | | 0.71 (0.47, 1.08) | 0.11 |
| Richest | | | | | | | 0.55 (0.35, 0.88) | 0.01 |
| Adjusted R² ** | 0.31 | | 0.41 | | 0.19 | | 0.29 | |

Notes: HCA1: Human capital approach where hours spent was calculated using minimum wage rate for all and household income as denominator; OR: Odds Ratio; CI: Confidence Interval; Blanks indicate Not Applicable;

* One patient among general population and 24 patients among tea garden families did not incur any direct cost, hence, were excluded from this analysis;

**Cox and Snell $R^2$

(NTEP) of the country for designing social protection measures for the TB patients in India. To the best of our knowledge, our study followed the largest cohort of notified DS-TB patients from different groups followed till date throughout their TB treatment period and beyond and study participants were representative of their respective groups.

Our study showed at least 30% of TB affected households faced catastrophic cost using HCA method of indirect cost calculation. However, that proportion was as high as 61% using another indirect cost calculation method (OA). As HCA method of indirect cost calculation

**Table 5. Likelihood of incurring catastrophic cost using HCA2 method of indirect cost calculation.**

| | General population (N = 528) | | Urban slum dwellers (N = 526) | | Tea garden families (N = 403) | | All participants (N = 1457) | |
|---|---|---|---|---|---|---|---|---|
| Explanatory variables | Adjusted OR (95% CI) | p-value | Adjusted OR (95% CI) | p-value | Adjusted OR (95% CI) | p-value | Adjusted OR (95% CI) | p-value |
| **Age (in years)** | 1.02 (1.00, 1.04) | 0.06 | 1.01 (0.99, 1.03) | 0.27 | 1.02 (1.00, 1.03) | 0.12 | 1.01 (1.00, 1.02) | 0.04 |
| **Sex** | | | | | | | | |
| Male (Reference) | 1.00 | | 1.00 | | 1.00 | | 1.00 | |
| Female | 1.18 (0.65, 2.14) | 0.58 | 1.64 (0.94, 2.86) | 0.08 | 1.58 (0.98, 2.56) | 0.06 | 1.49 (1.11, 2.02) | 0.01 |
| **Education** | | | | | | | | |
| Up to primary education (Reference) | 1.00 | | 1.00 | | 1.00 | | 1.00 | |
| Secondary education | 1.07 (0.56, 2.06) | 0.84 | 0.65 (0.35, 1.12) | 0.16 | 1.29 (0.68, 2.44) | 0.43 | 0.80 (0.57, 1.12) | 0.19 |
| Higher secondary education and above | 0.48 (0.22, 1.04) | 0.06 | 0.60 (0.27, 1.34) | 0.21 | 0.54 (0.20, 1.45) | 0.22 | 0.62 (0.40, 0.98) | 0.04 |
| **Pre-TB annual household income (Indian Rupee)** | | | | | | | | |
| Less than 100,000 (Reference) | 1.00 | | 1.00 | | 1.00 | | 1.00 | |
| 100,000 –less than 200,000 | 0.06 (0.03, 0.13) | <0.001 | 0.15 (0.08, 0.28) | <0.001 | 0.21 (0.12, 0.36) | <0.001 | 0.14 (0.10, 0.19) | <0.001 |
| 200,000 and above | 0.01 (0.00, 0.02) | <0.001 | 0.03 (0.01, 0.06) | <0.001 | - - - | | 0.03 (0.02, 0.06) | <0.001 |
| **Health insurance** | | | | | | | | |
| Having health insurance (Reference) | 1.00 | | 1.00 | | 1.00 | | 1.00 | |
| Not having health insurance | 1.22 (0.65, 2.30) | 0.54 | 1.03 (0.53, 1.99) | 0.93 | 1.00 (0.60, 1.66) | 1.00 | 1.02 (0.74, 1.41) | 0.90 |
| **Type of TB** | | | | | | | | |
| Pulmonary TB (Reference) | 1.00 | | 1.00 | | 1.00 | | 1.00 | |
| Extrapulmonary TB | 1.54 (0.82, 2.91) | 0.18 | 2.12 (1.18, 3.81) | 0.01 | 1.29 (0.76, 2.21) | 0.35 | 1.71 (1.24, 2.35) | 0.00 |
| **Delay from symptom initiation to treatment** | 1.02 (0.99, 1.05) | 0.22 | 1.01 (0.98, 1.04) | 0.63 | 1.01 (0.98, 1.04) | 0.41 | 1.01 (1.00, 1.03) | 0.08 |
| **Direct cost of TB treatment (Log cost)** * | 11.87 (7.23, 19.48) | <0.001 | 7.48 (4.91, 11.37) | <0.001 | 1.98 (1.60, 2.45) | <0.001 | 4.00 (3.33, 4.77) | <0.001 |
| **Residential status** | | | | | | | | |
| Urban (Reference) | 1.00 | | - - - | | - - - | | 1.00 | |
| Rural | 0.69 (0.38, 1.24) | 0.21 | | | | | 0.98 (0.71, 1.35) | 0.89 |
| **Wealth quintile** | | | | | | | | |
| Poorest (Reference) | - - - | | - - - | | - - - | | 1.000 | |
| Poorer | | | | | | | 0.66 (0.43, 1.04) | 0.07 |
| Middle | | | | | | | 0.63 (0.41, 0.99) | 0.04 |
| Richer | | | | | | | 0.73 (0.47, 1.14) | 0.16 |
| Richest | | | | | | | 0.40 (0.24, 0.67) | <0.001 |
| Adjusted $R^2$ ** | 0.47 | | 0.41 | | 0.20 | | 0.35 | |

Notes: HCA2: Human capital approach where hours spent was calculated using combination of patient wage and minimum wage and household income as denominator; OR: Odds Ratio; CI: Confidence Interval; Blanks indicate Not Applicable;

* One patient among general population and 24 patients among tea garden families did not incur any direct cost, hence, were excluded from this analysis;

**Cox and Snell $R^2$

put a value on people's time regardless of whether it is associated with monetary income, our main method of indirect cost calculation was HCA1 where we put equal value for time losses for patients and guardians as it considers the equity angle. However, we also used two other approaches to examine the differences in total cost using different methods of indirect cost calculation. For patients among general population, treatment cost was higher using HCA2 (INR 34,315) as compared to HCA1 (INR 32,829) because on average reported patient wage was higher than the minimum wage of the respective states. In contrast, treatment cost was much

**Table 6. Likelihood of incurring catastrophic cost using OA1 method of indirect cost calculation.**

| Explanatory variables | General population (N = 528) | | Urban slum dwellers (N = 526) | | Tea garden families (N = 403) | | All participants (N = 1457) | |
|---|---|---|---|---|---|---|---|---|
| | Adjusted OR (95% CI) | p-value | Adjusted OR (95% CI) | p-value | Adjusted OR (95% CI) | p-value | Adjusted OR (95% CI) | p-value |
| Age (in years) | 0.99 (0.98, 1.01) | 0.37 | 1.00 (0.99, 1.02) | 0.73 | 1.01 (1.00, 1.03) | 0.13 | 1.00 (0.99, 1.01) | 0.58 |
| **Sex** | | | | | | | | |
| Male (Reference) | 1.00 | | 1.00 | | 1.00 | | 1.00 | |
| Female | 0.41 (0.27, 0.63) | <0.001 | 0.71 (0.47, 1.06) | 0.09 | 0.67 (0.43, 1.03) | 0.08 | 0.57 (0.45, 0.73) | <0.001 |
| **Education** | | | | | | | | |
| Up to primary education (Reference) | 1.00 | | 1.00 | | 1.00 | | 1.00 | |
| Secondary education | 0.63 (0.39, 1.02) | 0.06 | 0.78 (0.50, 1.22) | 0.28 | 1.17 (0.65, 2.11) | 0.59 | 0.77 (0.58, 1.02) | 0.06 |
| Higher secondary education and above | 0.49 (0.29, 0.83) | 0.01 | 0.73 (0.41, 129) | 0.27 | 0.80 (0.34, 1.85) | 0.60 | 0.69 (0.49, 0.98) | 0.04 |
| **Pre-TB annual household income (Indian Rupee)** | | | | | | | | |
| Less than 100,000 (Reference) | 1.00 | | 1.00 | | 1.00 | | 1.00 | |
| 100,000 –less than 200,000 | 0.45 (0.26, 0.77) | 0.00 | 0.54 (0.33, 0.88) | 0.01 | 1.05 (0.67, 1.65) | 0.83 | 0.67 (0.51, 0.88) | 0.00 |
| 200,000 and above | 0.34 (0.20, 0.60) | <0.001 | 0.28 (0.16, 0.47) | <0.001 | 1.30 (1.12, 1.51) | <0.001 | 0.41 (0.29, 0.58) | <0.001 |
| **Health insurance** | | | | | | | | |
| Having health insurance (Reference) | 1.00 | | 1.00 | | 1.00 | | 1.00 | |
| Not having health insurance | 0.76 (0.49, 1.17) | 0.21 | 0.99 (0.60, 1.61) | 0.96 | 0.81 (0.51, 1.29) | 0.38 | 0.81 (0.62, 1.05) | 0.10 |
| **Type of TB** | | | | | | | | |
| Pulmonary TB (Reference) | 1.00 | | 1.00 | | 1.00 | | 1.00 | |
| Extrapulmonary TB | 0.71 (0.46, 1.10) | 0.13 | 0.85 (0.54, 1.31) | 0.46 | 1.57 (0.95, 2.59) | 0.08 | 1.02 (0.79, 1.32) | 0.87 |
| **Delay from symptom initiation to treatment** | 1.01 (0.98, 1.03) | 0.59 | 1.02 (1.00, 1.05) | 0.09 | 1.02 (0.99, 1.05) | 0.24 | 1.01 (1.00, 1.03) | 0.05 |
| **Direct cost of TB treatment (Log cost) *** | 2.05 (1.67, 2.51) | <0.001 | 1.76 (1.44, 2.14) | <0.001 | 1.30 (1.12, 1.51) | <0.001 | 1.60 (1.45, 1.77) | <0.001 |
| **Residential status** | | | | | | | | |
| Urban (Reference) | 1.00 | | - - - | | - - - | | 1.00 | |
| Rural | 1.07 (0.72, 1.59) | 0.75 | | | | | 1.17 (0.90, 1.51) | 0.24 |
| **Wealth quintile** | | | | | | | | |
| Poorest (Reference) | - - - | | - - - | | - - - | | 1.00 | |
| Poorer | | | | | | | 1.09 (0.76, 1.56) | 0.65 |
| Middle | | | | | | | 1.34 (0.94, 1.93) | 0.11 |
| Richer | | | | | | | 0.91 (0.63, 1.30) | 0.60 |
| Richest | | | | | | | 0.73 (0.49, 1.08) | 0.12 |
| Adjusted R² ** | 0.16 | | 0.13 | | 0.07 | | 0.11 | |

Notes: OA1: Output approach with household income as denominator; OR: Odds Ratio; CI: Confidence Interval; Blanks indicate Not Applicable;

* One patient among general population and 24 patients among tea garden families did not incur any direct cost, hence, were excluded from this analysis;

**Cox and Snell $R^2$

higher using HCA1 (INR 29,960) as compared to HCA2 (INR 22,981) for patients from tea garden areas implying that their wage rate was much lower than minimum wage rate. Low wage rate of the tea garden workers had been acknowledged in the literature [17] and our findings were in line with those. Treatment cost for patients among urban slum dwellers were similar using HCA1 (INR 30,782) and HCA2 (INR 30,806) implying that as majority of them were daily wage earners, they earned like minimum wage rate of the study states. Treatment cost was the highest using OA as the method of indirect cost calculation for all groups implying high rate of unemployment and hence, income loss because of TB disease.

**Table 7. Coping strategies and other economic consequences.**

| Coping strategies | General population | Urban slum dwellers | Tea garden families |
|---|---|---|---|
| Borrowing in IP, N (%) | 234 (44%) | 222 (42%) | 121 (28%) |
| Amount borrowed, mean (SD) | 19043 (24447) | 16817 (31704) | 7355 (11922) |
| Selling / mortgage in IP, N (%) | 66 (12%) | 79 (15%) | 44 (10%) |
| Amount received from sell/mortgage, mean (SD) | 22759 (29773) | 25257 (44292) | 9102 (23081) |
| Borrowing in CP, N (%) | 146 (29%) | 169 (34%) | 93 (23%) |
| Amount borrowed, mean (SD) | 16869 (26948) | 15534 (23007) | 5438 (8412) |
| Selling / mortgage in CP, N (%) | 62 (12%) | 77 (16%) | 54 (14%) |
| Amount received from sell/mortgage, mean (SD) | 26676 (30876) | 21049 (23726) | 6455 (6759) |
| Savings withdrawn during entire illness, N (%) | 101 (20%) | 89 (18%) | 97 (24%) |
| Amount withdrawn, mean (SD) | 21969 (36321) | 14111 (17537) | 8103 (13129) |
| **Other economic consequences** | | | |
| **Had to do the following activities during TB treatment period** | **General population** | **Urban slum dwellers** | **Tea garden families** |
| Cut down consumption level for other household members, N (%) | 88 (18%) | 137 (28%) | 62 (16%) |
| Another household member started working, N (%) | 32 (6%) | 31 (6%) | 7 (2%) |
| Run up account in shop, N (%) | 72 (14%) | 81 (16%) | 73 (18%) |
| Withdrawn children from school / private tuition | 18 (4%) | 25 (5%) | 10 (3%) |
| Move to lower rent accommodation | 1 (0.2%) | 10 (2%) | NA |
| Used multiple strategies | 47 (9%) | 71 (14%) | 31 (8%) |
| **Unable to do the following activities during TB treatment period** | | | |
| Could not pay electricity /mobile / gas / cable bills | 161 (32%) | 192 (39%) | 116 (29%) |
| Could not buy medicines for other members / diseases | 36 (7%) | 51 (10%) | 7 (2%) |
| Could not pay school / tuition fees | 64 (13%) | 69 (14%) | 19 (5%) |
| Could not pay house / shop rent | 41 (8%) | 113 (23%) | NA |
| Could not contribution to family / social events | 132 (27%) | 146 (29%) | 67 (17%) |
| Used multiple options | 134 (27%) | 189 (38%) | 55 (14%) |

**Notes:** IP–intensive phase; CP–continuous phase; Number of participants in IP coping strategies–general population (529); urban slum dwellers (526); tea garden families (427); CP coping strategies and other economic consequences—general population (498); urban slum dwellers (495); tea garden families (396) because of loss-to-follow-up in CP; Average amounts for borrowing / sale / mortgage and savings were calculated only for those who reported using those strategies; NA–not applicable as tea garden families get accommodation from tea estates.

We used pre-TB annual household income and expenditure as denominators of catastrophic cost calculation. Studies have mentioned that household expenditure is a better measure of permanent income of the household as it generally stays stable over time [14, 15]. However, we noticed practical difficulties while collecting household expenditure data. First, the study questionnaire became lengthy with additional consumption related questions and both interviewers' and participants' fatigue were evident. Second, several participants were in younger age groups, were housewives or in older ages and were unaware of detailed household consumption expenditure. This led to another set of interviews with the household members with complete household expenditure knowledge. Therefore, even though household consumption expenditure is a better proxy, researchers need to decide whether it will be practical to collect the data. On the other hand, we did not notice hesitancy in reporting household income for most participants and we cross checked income data for different rounds of interviews and the reporting were consistent. As in this paper, we did not explore which indirect and/or catastrophic cost calculation method was strongly associated with long term outcomes, we are unable to recommend the best method. As treatment cost varied according to indirect cost calculation methods for

different patient groups, while calculating TB treatment cost, it would be ideal to estimate indirect cost using one method and present sensitivity analyses using other methods to get a range of costs as different methods of indirect cost calculations have different policy implications.

Our estimated TB treatment costs and proportion of households faced catastrophic costs for all three groups were much higher than estimated in a recent literature review in India [6]. Their estimated cost was lower probably because about half of their reviewed studies did not include EPTB patients; few studies reported costs till diagnosis or IP of treatment and most of the reviewed studies were small scale, cross-sectional. Our study reported much higher cost of EPTB treatment as compared to PTB treatment and estimated cost from symptom onset till treatment completion, hence, exclusion of EPTB and considering treatment cost till diagnosis or IP in reviewed studies underestimated total treatment cost.

For participants with PTB, 46%-50% of total treatment cost was incurred in the pre-treatment phase, while the same was about 68% for EPTB patients among general population and urban slum dwellers. Direct cost contributed 53%-58% of total cost incurred in pre-treatment phase. The reason of much higher pre-treatment cost for EPTB patients was expenses incurred in tests for confirming TB. As majority diagnosis tests required for EPTB patients (e.g., computed tomography scan, magnetic resonance imagining, biopsy) were not available in most government hospitals and patients relied on private laboratories for these tests, their pre-treatment cost increased significantly. Government needs to take steps to ensure that either these facilities are available in government hospitals, or these expensive tests can be offered to symptomatic TB patients at free of cost through private sector engagement strategy.

Another reason for high pre-treatment cost for all types of TB was because there was 7–9 weeks delay from TB symptom onset to treatment initiation which was much higher than generally accepted period of delay of 4 weeks [18, 19]. During this period, our study participants made several visits to different providers and spent money and time. Therefore, reduction in delay should be a policy priority during the process of TB elimination in the country as the delay not only resulted in increased economic burden on the patients and households but as per other study results, this probably led to increased severity of the disease, chance of death and risk of transmission in the society [20, 21]. The NTEP has been making continuous efforts to reduce the diagnosis delay by improving private sector engagement in TB care, implementing active case finding and using high-efficiency diagnosis tool [12]. Apart from these, the Programme should also emphasize on the demand side, i.e., awareness generation in the community around TB disease, free available diagnosis and treatment facilities and economic consequences of late diagnosis so that the symptomatic patients can choose the appropriate provider quickly which can reduce diagnosis delay, hence, cost of treatment. Recent national TB prevalence survey also emphasized the need for community awareness generation [22].

Our estimates of proportion of households faced catastrophic cost using HCA methods of indirect cost calculation (about 30%) were comparable with Philippines (28%) but lower than Myanmar (45%) DR Congo (56%) and Timor-Leste (85%) [23–26]. Using OA as the indirect cost calculation method, our estimates (61%-65%) were higher than Kenya (27%); but were comparable with Vietnam (60%); Myanmar (60%); Lao PDR (63%) and Ghana (64%) [25, 27–30]. Further, it should be noted that for all these countries, catastrophic cost calculations included patients with multi-drug resistant TB while our estimates were only for DS-TB patients. Therefore, not only the disease burden is high in India, but it also has severe economic consequences. Unemployment and income loss related to TB in India is a major area of concern as was evident from high proportion of catastrophic cost using OA and policies need to be developed to ensure that TB care does not disrupt livelihood of the patients. It should be

noted in this context that though all these countries including ours followed the costing methodology suggested by WHO [2], the differences in the estimates across different countries could be because we followed up a cohort while other studies were cross-sectional and because of differences in wage rates, study settings and health systems.

Even though we estimated indirect cost, total cost and catastrophic cost associated with TB treatment in India using different approaches following 1482 notified DS-TB patients, the limitations of the study merit comment. First, as per sample size calculation, we required 512 notified DS-TB patients in each group, i.e., total 1536 patients to estimate the mean treatment cost with 95% confidence. We were able to reach out to 529 patients among general population and 526 among urban slum dwellers, however, we could not reach the required number of patients in tea garden areas. We interviewed 427 patients from 182 tea gardens across 6 districts. Given the time and budget of the study, it was not possible to add more districts / tea gardens, therefore, the results for tea garden areas may not be as precise as we expected. However, as the districts we covered in our study had maximum tea gardens, we expect our study participants were representative of their group. Second, household asset ownership could have been another way to estimate the capacity to pay for the TB affected household and that could have been used as denominator in catastrophic cost calculation [2]. However, that requires a national dataset with asset scoring. We were unable to find such dataset in Indian context, hence, could not calculate capacity to pay using household asset ownership. Third, our study participants were notified DS-TB patients who sought treatment from government health facilities. Therefore, our estimates did not cover patients who were not notified, sought treatment from private providers and had drug-resistant TB. Therefore, we expect that our estimated costs and proportion faced catastrophic costs were underestimated. Finally, patients were asked to retrospectively report treatment cost and income which were subject to recall bias, however, those were minimized by reviewing the relevant medical records wherever available and by interviewing patients twice over the course of treatment.

## Conclusions

Our study has important policy implications. One of the major contributors of catastrophic cost was the delay incurred from onset of symptoms till treatment initiation (ranging from 44% to 59% of total treatment cost). This delay can be reduced by ensuring early case detection through intense engagement with private providers as 71%-75% of our study participants from general population and urban slum dwellers first visited private providers after symptom onset and through active case finding. These are all supply side initiatives; however, the time has come to focus on the demand side as well. Awareness generation among the community may play a significant role in this context. Earlier advocacy, communication and social mobilization activities by the government and partners were not very successful because of insufficient intensity, duration, and scale. Therefore, interventions need to be developed to help community identifying TB symptoms quickly and seek appropriate pathway of care. All these efforts together can only reduce a significant proportion of treatment cost and hence, catastrophic cost. Further, reimbursement of expenses incurred in the pre-treatment phase through health insurance can also protect TB patients from the devastating economic consequences. However, our study showed poor health insurance coverage and limited usage among study participants, therefore, improved insurance coverage and reimbursement of expenses through insurance can play a crucial role in reducing catastrophic cost. Finally, policies need to be developed to protect TB patients from unemployment and income loss because of the disease which clearly was a major reason of financial hardship.

## Supporting information

**S1 Table. Descriptive statistics of explanatory variables in regression analysis.**
(DOCX)

**S2 Table. Likelihood of incurring catastrophic cost using HCA1 method of indirect cost calculation.**
(DOCX)

**S3 Table. Likelihood of incurring catastrophic cost using HCA2 method of indirect cost calculation.**
(DOCX)

**S4 Table. Likelihood of incurring catastrophic cost using OA1 method of indirect cost calculation.**
(DOCX)

**S1 Data. Complete dataset.**
(ZIP)

## Acknowledgments

We express our sincere gratitude to the Central TB Division, study state and district TB officers, senior treatment supervisors and health visitors for their tremendous support throughout the study period. We are thankful to Dr Arpita Ghosh for helping with the sampling strategy. We are grateful to Aparna K, Biswajit Dutta, Dipu Chakrabarty, Lily Rengma, Purbali Tungkhungia, Risenga Sebu Rengma, Ritul Das, Sakshi Rane, Suchismita Mullick, Swetha Ramesh and Vivek A for their excellent support during field work. Finally, we are ever indebted to all our study participants for taking part in our study and patiently sharing their experiences with us despite poor health conditions.

## Author Contributions

**Conceptualization:** Anna Vassall.

**Data curation:** Susmita Chatterjee, Palash Das, Gayatri Bhambure.

**Formal analysis:** Susmita Chatterjee, Palash Das, Guy Stallworthy, Gayatri Bhambure.

**Funding acquisition:** Susmita Chatterjee.

**Investigation:** Susmita Chatterjee.

**Methodology:** Susmita Chatterjee, Guy Stallworthy, Anna Vassall.

**Project administration:** Susmita Chatterjee, Palash Das, Radha Munje.

**Resources:** Susmita Chatterjee.

**Supervision:** Susmita Chatterjee, Palash Das, Radha Munje, Anna Vassall.

**Validation:** Susmita Chatterjee, Palash Das, Anna Vassall.

**Visualization:** Susmita Chatterjee, Guy Stallworthy, Gayatri Bhambure, Anna Vassall.

**Writing – original draft:** Susmita Chatterjee.

**Writing – review & editing:** Palash Das, Guy Stallworthy, Gayatri Bhambure, Radha Munje, Anna Vassall.

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
