## [Decision Letter · Decision Letter 0]

5 Nov 2023

PGPH-D-23-02005

Catastrophic costs for tuberculosis patients in India: impact of methodological choices

Dear Dr. Chatterjee,

Thank you for submitting your manuscript to PLOS Global Public Health. After careful consideration, we feel that it has merit but does not fully meet PLOS Global Public Health’s publication criteria as it currently stands. Therefore, we invite you to submit a revised version of the manuscript that addresses the points raised during the review process.

We look forward to receiving your revised manuscript.

Kind regards,

Habib Hasan Farooqui, MBBS, MD

Academic Editor

Journal Requirements:

Additional Editor Comments (if provided):

Reviewers' comments:

Reviewer's Responses to Questions

**Comments to the Author**

1. Does this manuscript meet PLOS Global Public Health’s publication criteria? Is the manuscript technically sound, and do the data support the conclusions? The manuscript must describe methodologically and ethically rigorous research with conclusions that are appropriately drawn based on the data presented.

Reviewer #1: Partly

Reviewer #2: Yes

2. Has the statistical analysis been performed appropriately and rigorously?

Reviewer #1: Yes

Reviewer #2: No

3. Have the authors made all data underlying the findings in their manuscript fully available (please refer to the Data Availability Statement at the start of the manuscript PDF file)?

Reviewer #1: Yes

Reviewer #2: No

4. Is the manuscript presented in an intelligible fashion and written in standard English?

Reviewer #1: Yes

Reviewer #2: Yes

5. Review Comments to the Author

Reviewer #1: Manuscript number PGPH-D-23-02005

Title: Catastrophic costs for tuberculosis patients in India: impact of methodological choices

General comments: This research estimated the catastrophic expenditure due to TB. The authors tried to estimate through different indirect cost calculation methods. This manuscript addresses a very important issue on catastrophic cost due to TB and it was done in four states Assam, Maharashtra, Tamil Nadu, and West Bengal. Still it is not clear that which method is most appropriate to be used to calculate indirect cost from this study. Overall this is a good study, but the current version is not suitable for publication and I would recommend this version to have a major revision for being accepted for publication.

Specific comments

• I would recommend the authors to modify the abstract incorporating the comments given. The authors can modify the sentences which starts with numbers throughout the manuscript.

• The sentence in the abstract “Studies included in the review were small scale cross sectional, the results of which are difficult to use for policy decisions” has to rephrased. It can’t be emphasized that all small scale cross sectionals studies are difficult to use for policy decisions.

• The novelty of this study has not come out well from this current version of the manuscript. There is no mentioning of the salient findings and conclusion of the study.

• The last sentence in the introduction also needs to be rephrased.

• How the four states are considered to be representative of India.

• How the people from tea garden areas are more prone to TB; is it defined in the national strategic plan; please check.

• I recommend the analysis to be done separately for respective population; since urban slum and rural may have more catastrophic expenditure. It is not sampled separately sub group analysis can be done for belter understanding.

• The reason for using different methodologies and the appropriate use of methodologies have not been given.

• In terms of indirect cost, the reason for the loss of income is due to TB or something else? How this is collected? Sometimes, the loss of income may be due to unemployment. Further, when patients can go for work, but doctors’ advise them to take rest during the treatment. In this scenario, some might go for work while some may not. Is this accounted in the study?

• Self-reported income and expenditure was collected for this study. Is any quality control mechanism followed? (For example, any cross checking)

• The cost for death, loss to follow up and relapse will be high. The current results might be under estimated. This needs to be included in the limitation.

• For extra pulmonary TB, the diagnostic costs will be very high due to the lack of diagnostic tools. That needs to be discussed elaborately.

• Authors used two different terms, ‘Discontinuation of treatment ‘and ‘Loss to follow up’. Is this different or same?

• The current treatment in India is daily regimen and drugs are supplied to patients. Whether DOT is followed now?

• INR can be replaced by ₹.

• The current study changed the denominator for catastrophic cost calculation. How this can be compared with previous studies?

• What is the role of the current intervention in reducing the catastrophic expenditure like NIKSHAY MITRA, ₹500 per month to patients, active case finding rapid diagnostic tools and molecular diagnostic which diagnoses early before patients becomes very sick. Is there any role health insurance in reducing the catastrophic cost since majority have health insurance.

• The TB treatment is completely free. Why patients are borrowing during CP. The reasons can be given. Since it is expected, the patients can become better after IP that they can go for work. Is this borrowing is for treatment or something else?

• Is it possible to give the factors that contributes to catastrophic expenditure?

• In line 252: no trace to be rephrased.

• In line 340 to 341 needs clarity.

• In line 351,352, the sentence to be changed to start with words.

• In line 391-393, 398-400 and 436-437 needs to be rephrased with clarity.

• The conclusion needs to be rewritten

• It was not estimated the current study costs for MDR, Relapse and lost to follow-up patients; but it was mentioned in the conclusion. Conclusion form the current study evidences. Similarly for insurance and Nikshay Poshan Yojana.

• Reference to be followed the journal style. Reference number 2, 5, 12, 13 and 21 seems to be incomplete reference

• Figure axis name is to be uniform (currently, some figure title is in capital letters and some are in small letter)

Reviewer #2: The present study estimates the economic burden (comprising of direct and indirect costs) and associated financial catastrophe among households with a member taking treatment for drug susceptible TB. Two specific methodologies i.e., human capital and output approach, were used to estimate indirect costs. Further six different scenarios were explored to provide a range of households facing catastrophic costs depending upon the type of indirect cost calculation method and the denominator used for the calculation of catastrophic cost. I congratulate the authors in successfully undertaking this analysis. Although, overall, the study methodology and analysis looks great, I have a few comments as follows:

1. The authors have stated in the discussion that it is difficult to comment upon the best method for indirect cost calculation as different methods have different policy implications. I recommend discussing the pros and cons of each indirect cost calculation method. As the present analysis shows a wide range of households facing catastrophic cost, i.e., from 30% to 70%, based on the type of method used for indirect cost calculation, authors need to provide an overall conclusion that which specific approach (may be not the best) is more suitable for indirect cost calculation within the Indian context.

HCA measures the lost productivity but does not capture indirect cost due to reduced productivity (as it just multiplies the time spent while seeking and receiving care with the wage rate). While on the other hand, OA seems to capture the reduced productivity in the form of income loss, but it misses the indirect costs associated with the time spent by those guardians or carers who are involved in unpaid household services (such as housewives). As the authors have stated, I also believe that each approach tries to capture the different components of indirect costs. But, as the main highlight of the study is to assess catastrophic costs using different methods of indirect cost calculation, I suggest authors to include more comprehensive arguments (as above) related to these indirect cost methods.

2. Authors need to clearly state in the methods that whether both direct and indirect cost were included in the calculation of catastrophic costs, as it seems from table 1 that only indirect costs were included.

3. To have a clear picture of the treatment costs, please include a separate descriptive table, either in the main text or in the appendix, showing the component wise breakdown of the direct and indirect cost at different stages of the TB treatment, among the three different sub-groups studied in the present analysis.

4. Please add a title for y axis in fig. 4 and 5. Y axis seems to show the proportion of households facing financial catastrophe. In fig. 4, I would recommend providing a ‘p value’ and showing whether the difference in the extent of catastrophic costs across the income groups was statistically significant or not. Likewise, a ‘p-value’ for showing difference in catastrophic costs across type of TB in Fig. 5 could be mentioned.

5. I could not interpret and fully understand the relevance of Fig. 3. While some of the blue dots suggest that pre-TB annual household income is zero, and the households faced some amount of expenditure in the pre-treatment phase. I am unsure, I am correctly interpreting the figure, but how could annual household income be zero? Likewise, in Fig. 2, some of the dots correspond to a pre-TB annual household income of zero. Please provide more information and explanation in interpreting these figures.

6. The authors mentioned in Table 2 that around 20-30% of the households had some form of health insurance. Authors may provide more details on the type of insurance and which specific services were covered under it. Further, it would be great to see whether the presence of insurance, in comparison to households without any health insurance, led to any reduction in OOPE or catastrophic costs.

7. In addition to univariable analysis, I would suggest including a multiple regression to examine the association or risk of incurring catastrophic costs, with demographic characteristics, clinical characteristics, adverse events, treatment seeking behaviour, income status, residential status, presence of insurance, etc.

6. PLOS authors have the option to publish the peer review history of their article (what does this mean?). If published, this will include your full peer review and any attached files.

**Do you want your identity to be public for this peer review?** For information about this choice, including consent withdrawal, please see our Privacy Policy.

Reviewer #1: **Yes: **Malaisamy Muniyandi

Reviewer #2: No

---

## [Decision Letter · Decision Letter 1]

23 Jan 2024

PGPH-D-23-02005R1

Catastrophic costs for tuberculosis patients in India: impact of methodological choices

Dear Dr. Chatterjee,

Thank you for submitting your manuscript to PLOS Global Public Health. After careful consideration, we feel that it has merit but does not fully meet PLOS Global Public Health’s publication criteria as it currently stands. Therefore, we invite you to submit a revised version of the manuscript that addresses the points raised during the review process.

We look forward to receiving your revised manuscript.

Kind regards,

Habib Hasan Farooqui, MBBS, MD

Academic Editor

Journal Requirements:

2. Please send a completed 'Competing Interests' statement, including any COIs declared by your co-authors. If you have no competing interests to declare, please state "The authors have declared that no competing interests exist". Otherwise please declare all competing interests beginning with twhe statement "I have read the journal's policy and the authors of this manuscript have the following competing interests:"

Additional Editor Comments (if provided):

Dear Authors,

Thank you for revising the manuscript in light of the reviewers comments.

There is a suggestion from one of the reviewers with regards to estimating probability and magnitude of the catastrophic cost and its determinants. You may consider that as part of sensitivity analysis.

Please see attached reviewers comment.

Thank you,

Kind regards,

Reviewers' comments:

Reviewer's Responses to Questions

**Comments to the Author**

1. If the authors have adequately addressed your comments raised in a previous round of review and you feel that this manuscript is now acceptable for publication, you may indicate that here to bypass the “Comments to the Author” section, enter your conflict of interest statement in the “Confidential to Editor” section, and submit your "Accept" recommendation.

Reviewer #1: All comments have been addressed

Reviewer #2: (No Response)

2. Does this manuscript meet PLOS Global Public Health’s publication criteria? Is the manuscript technically sound, and do the data support the conclusions? The manuscript must describe methodologically and ethically rigorous research with conclusions that are appropriately drawn based on the data presented.

Reviewer #1: Yes

Reviewer #2: Yes

3. Has the statistical analysis been performed appropriately and rigorously?

Reviewer #1: Yes

Reviewer #2: Yes

4. Have the authors made all data underlying the findings in their manuscript fully available (please refer to the Data Availability Statement at the start of the manuscript PDF file)?

Reviewer #1: Yes

Reviewer #2: Yes

5. Is the manuscript presented in an intelligible fashion and written in standard English?

Reviewer #1: Yes

Reviewer #2: Yes

6. Review Comments to the Author

Reviewer #1: The authros addressed all th comments. teh current version is suitable for publication

Reviewer #2: Authors have appropriately addressed most of the comments, and I believe this manuscript is almost ready for publication. Nevertheless, I would like to propose a suggestion:

I acknowledge that authors have used "Proportion of total TB treatment cost in pre-TB annual household income" for assessing the risk of catastrophic expenditure in linear regression. However, may I suggest considering logistic regression model instead of linear regression, with the proportion of households facing catastrophic expenditure as the dependent variable. Logistic regression is also better suited for capturing the impact of independent categorical variables (with multiple categories) such as education, wealth quintiles, etc. This adjustment could provide a better understanding of the predictors influencing the risk of facing catastrophic expenditure.

7. PLOS authors have the option to publish the peer review history of their article (what does this mean?). If published, this will include your full peer review and any attached files.

**Do you want your identity to be public for this peer review?** For information about this choice, including consent withdrawal, please see our Privacy Policy.

Reviewer #1: **Yes: **Malaisamy Muniyandi

Reviewer #2: No

---

## [Decision Letter · Decision Letter 2]

19 Mar 2024

Catastrophic costs for tuberculosis patients in India: impact of methodological choices

PGPH-D-23-02005R2

Dear Dr Chatterjee, 

We are pleased to inform you that your manuscript 'Catastrophic costs for tuberculosis patients in India: impact of methodological choices' has been provisionally accepted for publication in PLOS Global Public Health.

Best regards,

Habib Hasan Farooqui, MBBS, MD

Academic Editor

Reviewer Comments (if any, and for reference):

Reviewer's Responses to Questions

**Comments to the Author**

1. If the authors have adequately addressed your comments raised in a previous round of review and you feel that this manuscript is now acceptable for publication, you may indicate that here to bypass the “Comments to the Author” section, enter your conflict of interest statement in the “Confidential to Editor” section, and submit your "Accept" recommendation.

Reviewer #2: All comments have been addressed

2. Does this manuscript meet PLOS Global Public Health’s publication criteria? Is the manuscript technically sound, and do the data support the conclusions? The manuscript must describe methodologically and ethically rigorous research with conclusions that are appropriately drawn based on the data presented.

Reviewer #2: Yes

3. Has the statistical analysis been performed appropriately and rigorously?

Reviewer #2: Yes

4. Have the authors made all data underlying the findings in their manuscript fully available (please refer to the Data Availability Statement at the start of the manuscript PDF file)?

Reviewer #2: Yes

5. Is the manuscript presented in an intelligible fashion and written in standard English?

Reviewer #2: Yes

6. Review Comments to the Author

Reviewer #2: None

7. PLOS authors have the option to publish the peer review history of their article (what does this mean?). If published, this will include your full peer review and any attached files.

**Do you want your identity to be public for this peer review?** For information about this choice, including consent withdrawal, please see our Privacy Policy.

Reviewer #2: **Yes: **Akashdeep Singh Chauhan
